# Preparation and Characterization of Polyvinylalcohol/Polysulfone Composite Membranes for Enhanced CO_2_/N_2_ Separation

**DOI:** 10.3390/polym15010124

**Published:** 2022-12-28

**Authors:** Ying Li, Danlin Chen, Xuezhong He

**Affiliations:** 1Department of Chemical Engineering, Guangdong Technion-Israel Institute of Technology, 241 Daxue Road, Shantou 515063, China; 2The Wolfson Department of Chemical Engineering, Technion-Israel Institute of Technology, Haifa 3200003, Israel

**Keywords:** polyvinyl alcohol, composite membranes, gas permeation, CO_2_ permeance, CO_2_/N_2_ selectivity

## Abstract

The unique properties of polyvinyl alcohol (PVA) and polysulfone (PSf), such as good membrane-forming ability and adjustable structure, provide a great opportunity for CO_2_-separation membrane development. This work focuses on the fabrication of PVA/PSf composite membranes for CO_2_/N_2_ separations. The membranes prepared by coating a 7.5 wt% PVA on top of PSf substrate showed a relatively thin selective layer of 1.7 µm with an enhanced CO_2_/N_2_ selectivity of 78, which is a ca. 200% increase compared to the pure PSf membranes. The CO_2_/N_2_ selectivity decreases at a rapid rate with the increase of feed pressure from 1.8 to 5 bar, while the CO_2_ permeance shows a slight reduction, which is caused by the weakening of coupling transportation between water and CO_2_ molecules, as well as membrane compaction at higher pressures. Increasing operating temperature from 22 °C to 50 °C leads to a slight decrease in CO_2_ permeance, but a significant reduction in the CO_2_/N_2_ selectivity from 78 to 27.1. Moreover, the mass transfer coefficient of gas molecules is expected to increase at a higher velocity, which leads to the increase of CO_2_ permeance at higher feed flow rates. It was concluded that the CO_2_ separation performance of the prepared membranes was significantly dependent on the membrane operating parameters, and process design and optimization are crucial to bringing CO_2_-separation membranes for industrial applications in post-combustion carbon capture.

## 1. Introduction

With the acceleration of industrialization, the emissions of CO_2_ and other greenhouse gases into the atmosphere have increased dramatically around the world, which causes a significant global warming issue. Carbon capture, utilization, and storage (CCUS) is considered one of the most promising ways to mitigate CO_2_ emissions [1]. Compared to the conventional separation techniques such as absorption, adsorption, and cryogenic distillation, membrane separation technology offers the advantages of lower energy consumption, flexible operation, smaller footprint, and less or no chemical requirement, which shows great potential for CO_2_ separation [2,3,4]. Different types of membrane materials such as dense polymers, mixed matrix membranes, facilitated transport membranes, and inorganic membranes have been developed for gas separations. Among them, inorganic membranes are usually highly costly [5], which limits their large-scale applications for CO_2_ capture from flue gas where a huge membrane area is usually required to process a high-volume flow of a gas stream, especially in power plants [6].

Polymeric membranes are much more available in industrial CO_2_ separations due to their relatively high gas permeability, easy assembling process, good chemical stability, and relatively low cost [7,8]. However, polymeric membranes are usually suffering from the trade-off between gas permeability and selectivity, which means high gas permeability will be usually accompanied by low selectivity and vice versa [9]. It was reported that none of the pure polymeric membranes can surpass the Robeson upper bound for the CO_2_-N_2_ separation pair [10]. To successfully implement the membrane technology for industrial CO_2_ capture, novel membrane materials should be developed to overcome this trade-off to bring down capital cost related to the required membrane area (determined by gas permeability) and the operating cost related to the energy demand [10,11]. Novel strategies should be developed to ensure that the membrane performance reaches the top-right corner of the upper bound plot. Although the upper bound is only applied originally to dense polymeric membranes, gas permeability and selectivity are still the two main parameters to characterize the separation performances for any membrane materials [12,13]. Various ways exhibit performance that exceeds the upper bound, such as mixed matrix membranes (MMMs) [14], hybrid membranes [15], and composite membranes. Among them, composite membranes show significant potential for enhanced CO_2_ separation performance, as is evident from their widespread applications in liquid or gas separations [16].

Polysulfone (PSf) membrane is widely used as polymeric support due to the advantages of low material cost, high mechanical strength, and good thermal and chemical stabilities [17]. Such material can form a porous structure with regular micropores, which is suitable for fabricating high-performance gas separation membranes [18]. Meanwhile, some literature reported that the hydrophilic polyvinyl alcohol (PVA) can form a dense selective layer to present a highly swelling behavior with good film-forming properties [19,20]. It is worth noting that pure PSf or PVA membranes cannot achieve high selectivity and permeance simultaneously [21,22], but the combination can significantly improve gas permeance while maintaining a high selectivity, as reported by Chung et al. [23]. Peng et al. [24,25] demonstrated that the PVA/PSf composite nanofiltration membranes fabricated by multi-step coating above PSf with a defect-free PVA layer showed improved membrane performance. Moreover, it was also reported that high-performance composite membranes can be formed by combining these two polymers [26], although the preparation of a completely defect-free membrane is still challenging. However, the incompatibility between the hydrophobic PSf substrate and the hydrophilic top layer may lead to delimitation [27,28], which lowers the membrane stability and lifetime and limits its application. In order to address this issue, some literature reported introducing a surfactant of sodium dodecyl sulfate (SDS) into the coating solution to enhance the adhesion and compatibility [29,30]. Although adding PVA selective layer can enhance the selectivity, the permeabilities of the penetrants were still low under dry conditions, as reported by Chao et al. [31], as the hydrogen bonds of the polymer chain hinder CO_2_ and N_2_ permeation [32]. Therefore, it is crucial to operate this type of membrane in a humidified condition [33] where the polymer chain can be swelled to enhance the gas diffusivity coefficient. It was reported that water molecules can break the hydrogen bonds between molecules and reduce the crystallinity of the PVA chain [34], and thus the CO_2_/N_2_ selectivity and CO_2_ permeability for the water-swollen PVA membranes were significantly improved, which is mainly caused by the enhanced gas diffusivity inside the polymer matrix (Permeability = diffusivity × solubility coefficient), while the increase of N_2_ diffusivity is not as significant as CO_2_ due to the competing transport, and thus leads to the increase of CO_2_/N_2_ selectivity as well [35].

However, a systematic investigation of the membrane preparation parameters and the optimal operating condition on the PVA/PSf composite membrane performances is still lacking. Therefore, in this work, the membrane preparation parameters such as PVA content and coating thickness were optimized to identify the best condition for fabricating high-performance composite membranes, and the process operating parameters for CO_2_/N_2_ separation were also systematically tested.

## 2. Materials and Methods

### 2.1. Materials

The PSf membranes (MWCO 20K, Alfa Laval, Shanghai, China) were used as support layers. PVA (MW 89,000–98,000, 99+%) was bought from ALDRICH Chemicals for the preparation of the coating solutions with deionized (DI) water. The adhesive SDS was purchased from Sigma-Aldrich (Shanghai, China). The mixed gas of 10 vol.% CO_2_/90 vol.% N_2_ and sweep gas of Argon (purity: 99.99%) were supplied from Chaozhou Dafeng Gas Co., Ltd. (Chaozhou, China) for gas permeation testing.

### 2.2. Fabrication of Composite Membranes

The PVA coating solutions with different polymer concentrations were prepared by dissolving a certain amount of PVA in DI water with magnetic stirring for 24 h to ensure the polymers were fully dissolved. The composition of the PVA coating solutions varies from 5–10 wt%. The prepared PVA solutions were filtered by 0.8 µm syringe filters to remove any impurity/dust and then put in the ultrasonic bath to remove air bubbles before coating. The composite membranes were made by coating a thin PVA selective layer on top of the PSf support. First, clean glass plates were used to fix the PSf support with the aluminum tap. The addition of the surfactant of SDS into the PVA solutions and the removal of the residual glycerol on the PSf surface by washing with warm tap water of 40–45 °C for at least one hour was conducted to enhance the compatibility between the hydrophobic PSf surface and hydrophilic PVA solution. Once the pre-treatment was completed, the PSf supports were gently dried in the atmosphere, which ensures the membrane is completely flat to achieve a uniform selective layer without any defects. It should be noted that the pre-treated supports should not be dried in an oven to avoid any significant shrinkage or wrinkle on the membrane surface. After that, placing the support on the coating machine to fabricate composite membranes, the coating bar with a small thickness of 20 µm was used to coat the PVA selective layer at a coating speed of 18 m/h. As it is shown in Figure 1, the coated membranes were dried in a connective oven at 45 °C overnight and followed by a heat treatment at 100 °C for 2–3 h to make composite membranes for CO_2_/N_2_ separations.

### 2.3. Membrane Characterization

The morphology and structure of the prepared composite membranes were characterized by scanning electron microscope (SEM) using a Zeiss GeminiSEM 450. Fourier-transform infrared (FTIR) spectra for the membrane samples were obtained by a Thermo scientific spectrometer (Thermo Nicolet iN10, Thermo Fisher Scientific Inc., Waltham, USA) by scanning the wavenumbers ranging from 500 to 4000 cm^−1^. The hydrophilicity of the prepared membranes was determined by the contact angle measurement with a Drop Shape Analyzer (KRUSS, DSA25, KRÜSS GmbH, Hamburg, Germany). The swelling degree (SD) of the prepared PVA/PSf composite membranes was evaluated by the weight ratio between the dried and wetted membranes [36]. The membrane samples were dried in a vacuum oven (45 °C) for at least 24 h and weighed immediately. After that, the membranes were placed in a closed saturated water vapors container to get fully humidified. By removing the surface water on the samples with tissue paper, the wetted membranes were weighed again.
(1)$$SD=\frac{W_{s}-W_{d}}{W_{d}}\times 100\%$$
where *W_s_* and *W_d_* are the weight (g) of the swelling membrane sample at saturation and the weight of a dried membrane, respectively.

### 2.4. Membrane Performance Testing for CO_2_/N_2_ Separation

The membrane performances for CO_2_/N_2_ separation were tested by a gas permeation rig equipped with a plate-and-frame membrane module with an effective area of 17 cm^2^ as indicated in Figure 2. A pre-mixed gas of 10 vol% CO_2_ and 90 vol% N_2_ was fed into the membrane module at a given flow rate controlled by a mass flow controller (MFC, Bronkhorst, Ruurlo, The Netherlands). The relative humidity (RH, fully humidified) of feed gas was controlled by passing the gas through a humidifier equipped with heating jacket. The feed and permeate lines and the module can be heated to the desired testing temperature by heating trace. Argon was used as a sweep gas to create the driving force to dilute the penetrated gases, and to ensure that the permeate gas continuously flowed to gas chromatography (GC, Shimadzu GC-2014, Shimadzu, Kyoto, Japan) with a thermal conductivity detector (TCD) for gas composition measurements. The feed and permeate pressures were controlled by two back-pressure controllers (BPC, Alicat, Alicat Scientific, Tucson, AZ, USA). It is worth noting that feed pressure of 2 bar is widely used in real-life applications due to the expected significantly increased power demand for the flue gas compression to be operated at a higher feed pressure. Therefore, most testing experiments were conducted at the feed pressure of 2 bar, except for the influence of feed flow rate, where feed pressure of 3 bar was used. The process operating parameters such as feed pressure, temperature, and gas flow rate were systematically investigated to document their influences on the separation performances of the developed membranes. The gas permeance *P_i_* is calculated by Equation (2) as shown below [37]:(2)$$P_{i}=\frac{q_{i}}{A\Delta p_{i}}$$
where *P_i_* is the gas permeance of component *i* expressed in Gas Permeance Units (GPU), 1 GPU = 10^−6^ cm^3^ (STP) cm^−2^ s^−1^ cm Hg^−1^ = 2.736 × 10^−3^ m^3^(STP) m^−2^ h^−1^ bar^−1^ = 3.35 × 10^−10^ mol m^−2^ s^−1^ Pa^−1^; *q_i_* is the permeate flow rate of gas species *i*, m^3^ (STP) h^−1^, which was measured by a digital mass flow meter (MFM, Bronkhorst, Ruurlo, The Netherlands) installed on the permeate line. *A* is the effective membrane area (1.7 × 10^−3^ m^2^), and $\Delta p_{i}$ is the partial pressure difference of *i* between the feed and permeate side (bar).

The selectivity (S) is calculated by Equation (3) as shown below:(3)$$S=\frac{y_{i}/y_{j}}{x_{i}/x_{j}}$$
where *y* and *x* are the mole fractions of gas components *i* and *j* in the permeate and feed sides, respectively.

## 3. Results and Discussion

### 3.1. Membrane Swelling Behavior

PVA is a hydrophilic membrane material and can be swollen in a humidified condition. Figure 3a shows that all of the prepared composite membranes present a high SD of ca. 70%. However, the difference between them is very small, which indicates that coating an extra PVA layer may not affect the properties of the PSf support.

Compared with the pure PSf membrane, the water contact angle of the PVA/PSf composite membranes reduces to around 37.5°, which is mainly caused by the hydrophilicity of the PVA layer and the addition of strong surface tension material (i.e., SDS). However, with the increase of the PVA concentration in the coating solutions, the membrane thickness increases (see the SEM images below), which leads to the reduction of the SDS contribution to the hydrophilicity of the whole membranes, and thus the decrease of the hydrophilicity, as shown in Figure 3b.

### 3.2. FTIR Spectra

Based on the FTIR spectra in Figure 4, it can be found that two strong characteristic peaks were observed at the wavelengths of 1489 and 1586 cm^−1^ for the pure PSf substrate [38], which indicates an aromatic vibrational bonding of C=C in the polysulfone group. Moreover, the peaks at the wavenumbers of 1151 and 1242 cm^−1^ are the vibrational bonding of O=S=O and C–O–C in the ether group. The wavenumber of 3056 cm^−1^ corresponds to the vibrational bonding of =C–H in the aromatic ring of polysulfone, and the two absorption peaks observed at the wavenumbers of 1295 and 1324 cm^−1^ represent the vibrational region of the sulfone group.

The large bands observed between 3000 and 3600 cm^−1^ are linked to the stretching O–H from the intermolecular and intramolecular hydrogen bonds [36]. The vibrational band observed between 2800 and 3000 cm^−1^ refers to the stretching C–H from alkyl groups. It can be concluded that the PVA selective layer was successfully coated on top of the PSf substrate.

### 3.3. Membrane Morphology and Structure

Figure 5 shows both surface and cross-section SEM images of different concentration membrane samples at a magnification of 1000× and 6000×, respectively. According to the surface images, the prepared membranes are smooth, defect-free, and homogenous. Moreover, as can be seen from the cross-section images, the layered structural characteristic of all membranes can be identified, and the thickness of the coating layers is between 1.2 to 5 μm.

### 3.4. CO_2_/N_2_ Mixed Gas Separation Performance

#### 3.4.1. Effect of the PVA Concentration

It can be found that gas permeance and selectivity of PVA/PSf composite membranes increase with the increase of the PVA concentration in the coating solutions as shown in Figure 6. It is suspected that the water-swollen PVA selectivity layer may seal the surface defects more thoroughly [33]. It is remarkable that the highest selectivity in the concentration of 7.5 wt% PVA indicates that the prepared membranes form a defect-free selective layer. However, a further increase in the polymer concentration may lead to the formation of a denser layer, and thus reduce the CO_2_ and N_2_ transportation through the membranes, which results in the decrease of both CO_2_ permeance and CO_2_/N_2_ selectivity. Therefore, 7.5 wt% was identified as the best condition for making PVA/PSf composite membranes, which were further tested for CO_2_/N_2_ separations at different operating conditions. It should be noted that the composite membranes present lower gas permeance compared to the pure PSf membranes (washed with tap water), as the extra PVA layer causes additional transport resistance [39,40]. Moreover, lower molecular weight PVA fractions might penetrate the PSf matrix to decrease the pore size of the substrate or cause pore blocking and thus leading to the decrease of gas permeance. Future work on seeking better support materials with higher gas permeances should be pursued to enhance the membrane performance.

#### 3.4.2. Effect of Feed Flow Rate

The feed flow rate, varying from 100–500 NmL/min with a constant sweep gas flow of 20 NmL/min under the feed pressure and temperature of 3 bar and 25 °C, respectively, was conducted to investigate its influence on the separation performance for the 7.5 wt% PVA/PSf membrane. It should be noted that it is difficult to get the higher flow rate of >300 NmL/min when the feed pressure is controlled at 2 bar by a back-pressure controller. Therefore, the feed pressure of 3 bar was applied for the variation of the feed flow rate, and the results are shown in Figure 7. It can be seen that CO_2_ permeance increases from 2.3 to 5.7 GPU with the increase of the feed flow rate. It should be noted that the gas velocity that passes through the given cross-sectional area of the membrane module increases at higher feed flow rates. Therefore, the mass transfer coefficient of gas molecules is expected to increase at a higher velocity [41], hence increasing the gas permeance. However, it is worth noting that both CO_2_ and N_2_ permeances will increase simultaneously as the mass transfer enhancement contributes to all gas molecules. Thus, the CO_2_/N_2_ selectivity does not present a significant change, as indicated in Figure 7. Further increasing the feed flow may enhance the gas permeance if a stable operation can be achieved at the given pressure. However, the feed gas to be processed by a specific membrane area (i.e., the ratio between feed flow and membrane area) is designed in the real-life application, and a membrane system is usually operated at a relatively high stage-cut. It is expected that the commercial modules should be operated at a low or moderate feed flow. Therefore, much higher feed flow rates of >600 NmL/min have not been conducted in this work.

#### 3.4.3. Effect of Operating Temperature

The gas permeation tests were conducted at different operating temperatures for the best membranes prepared by coating a 7.5 wt% PVA solution. The system was tested at the constant feed and sweep flow rates of 300 and 20 NmL/min with a feed pressure of 2 bar, and the results are shown in Figure 8. When the temperature increases from 22 to 50 °C, the CO_2_ permeance decreases slightly, but the CO_2_/N_2_ selectivity decreases significantly from 78.0 to 27.1. It is expected that the gas diffusivity will increase at higher operating temperatures due to kinetic domination [41]. However, from the thermodynamic point of view, the CO_2_ solubility coefficient decreases on the contrary. Overall, the CO_2_ permeance decreases with the increase in operating temperature, which means that this type of membrane might be more suitable for low or moderate operating temperatures. Moreover, the N_2_ solubility coefficient will not be significantly influenced by temperature due to its less condensability, so the N_2_ permeance slightly increases with the increase of temperature as the diffusivity dominates the N_2_ transportation through the membranes. Therefore, it presents a slight decrease in CO_2_/N_2_ selectivity at higher temperatures. However, it is worth noting that water vapor plays a significant role in the swelling of the membrane. Due to the limitation related to the equipment design, stable operation at high relative humidity (RH) cannot be achieved at higher operating temperatures. This should be further investigated in future work to split the influence of RH and temperature.

#### 3.4.4. Effect of Feed Pressure

Even though CO_2_ capture from flue gas is usually operated at relatively low feed pressure to avoid a significantly high energy consumption for the feed gas compression, it is worth investigating the membrane performance at different pressures to balance the operating cost related to power demand and the capital cost of the membrane unit. Therefore, the feed pressure variation from 1.8 to 5 bar was conducted under the constant feed and permeate flow rates of 300 and 20 NmL/min at 25 °C. The obtained membrane separation performances of CO_2_ permeance and CO_2_/N_2_ selectivity are shown in Figure 9. It can be seen that the CO_2_/N_2_ selectivity decreases at a rapid rate with feed pressure, while the CO_2_ permeance shows a slight reduction. It is expected that the absolute water vapor content at high pressures decreases, which leads to the weakening of coupling transportation between water and CO_2_ molecules. Moreover, high feed pressure causes the compacting of the membrane matrix to reduce the swelling effect. These two factors contribute to the reduction of CO_2_ permeation together. However, due to the decrease in membrane swelling, the competing transport between the N_2_ and CO_2_ molecules is expected to become more significant, which leads to a great reduction in the CO_2_/N_2_ selectivity. Therefore, it can be concluded that a relatively low feed pressure operation is preferable for the prepared PVA/PSf composite membranes to achieve a high separation performance for CO_2_ capture from flue gas.

## 4. Conclusions

PVA/PSf composite membranes were successfully prepared by coating a selective layer on top of PSf substrate. The composite membranes presented better separation performance compared to the pure PSf membranes, which indicates that coating PVA on PSf can effectively improve the membrane selectivity compared to the pure PSf support membranes. Under the same condition, a 7.5 wt% PVA coating solution is more suitable to achieve high performance, with a slightly lower CO_2_ permeance 3.4 GPU and 78.2 CO_2_/N_2_ selectivity, based on the results of the influences of polymer concentration on the membrane separation performance. As for the feed flow rate test, the membrane experiences a significant increase in CO_2_ permeance from 2.3 GPU at 100 NmL/min to 5.7 GPU at 500 NmL/min, while the CO_2_/N_2_ selectivity remains stable with slight upward redundancy, which is mainly caused by the enhanced mass transfer coefficient. With the increase of operating temperature, CO_2_ permeance decreases due to the significant reduction from the dominating parameter of the CO_2_ solubility coefficient. Thus, this type of membrane may be more suitable for low or moderate operating temperatures. Moreover, the composite membranes cannot maintain good separation performance under high-pressure operation due to the weakening of coupling transportation between water and CO_2_ molecules, as the absolute water vapor content decreases at higher feed pressures. In addition, high feed pressure causes membrane compaction to reduce the swelling effect. Therefore, process design and optimization are very important for bringing such materials into real-life industrial CO_2_ capture.

## Figures and Tables

**Figure 1 polymers-15-00124-f001:**
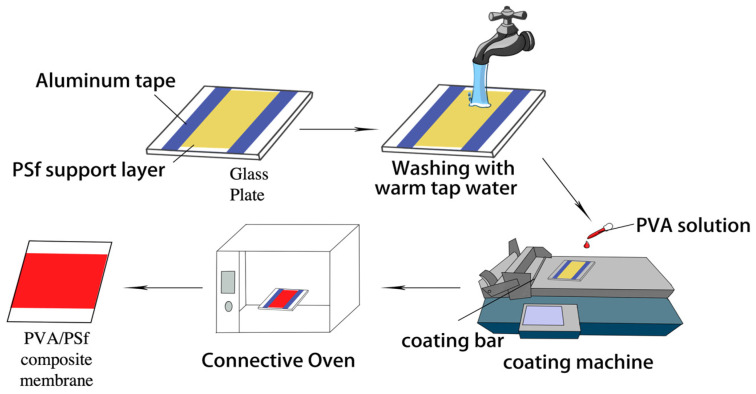
The fabrication procedure for PVA/PSf composite membranes.

**Figure 2 polymers-15-00124-f002:**
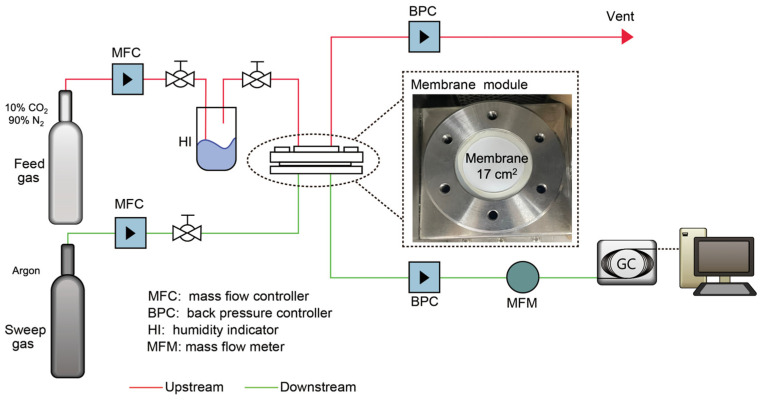
Schematic diagram of gas permeation rig for membrane performance testing.

**Figure 3 polymers-15-00124-f003:**
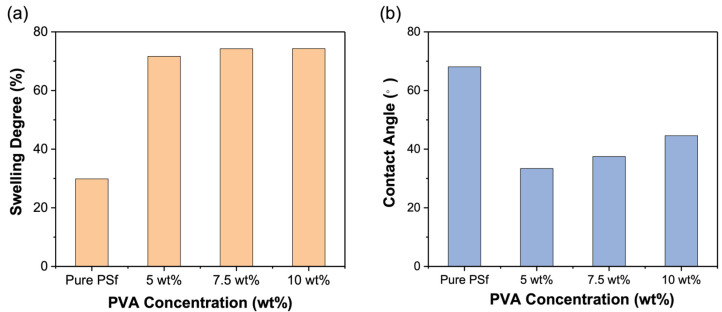
(**a**) The swelling degree and (**b**) the contact angle of pure PSf and PVA/PSf composite membranes.

**Figure 4 polymers-15-00124-f004:**
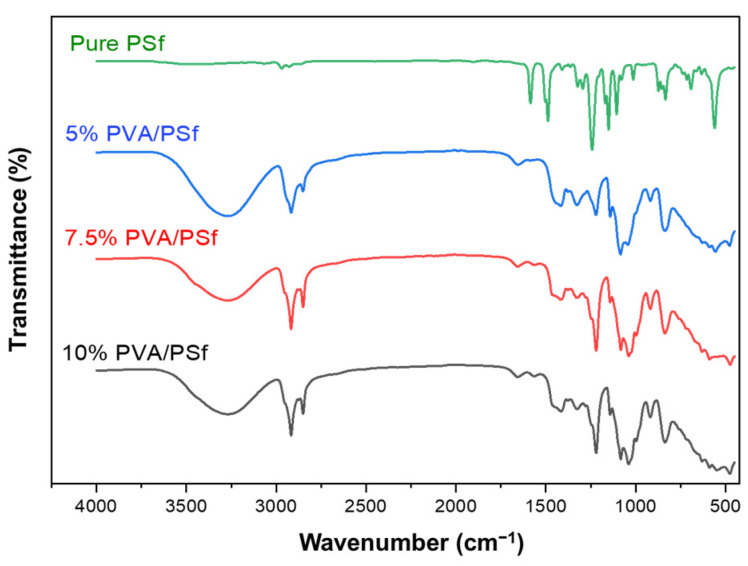
FTIR spectra of pure PSf and various PVA/PSf composite membranes.

**Figure 5 polymers-15-00124-f005:**
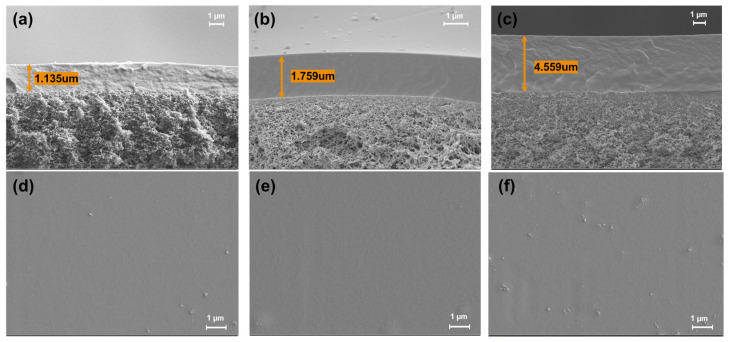
(**a**,**d**) Cross-sectional and surface images of 5 wt% PVA/PSf membrane; (**b**,**e**) Cross-sectional and surface images of 7.5 wt% PVA/PSf membrane; (**c**,**f**) Cross-sectional and surface images of 10 wt% PVA/PSf membrane.

**Figure 6 polymers-15-00124-f006:**
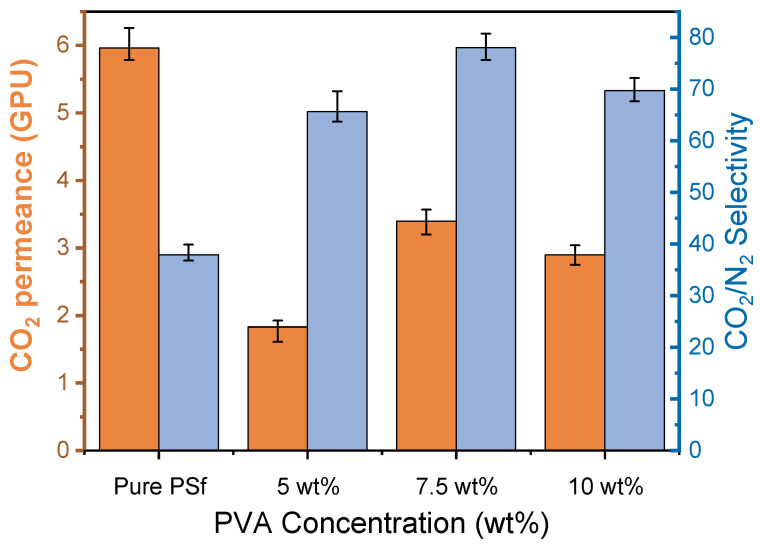
Separation performance of PVA/PSf membranes with different PVA concentration, tested with a feed flow rate of 300 NmL/min at 2 bar and 25 °C.

**Figure 7 polymers-15-00124-f007:**
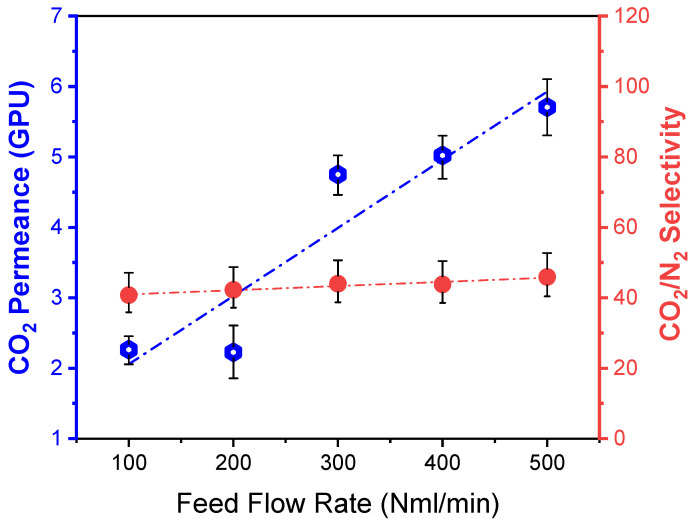
Dependence of the membrane performances on the feed flow rate under a feed pressure and a temperature of 3 bar and 25 °C.

**Figure 8 polymers-15-00124-f008:**
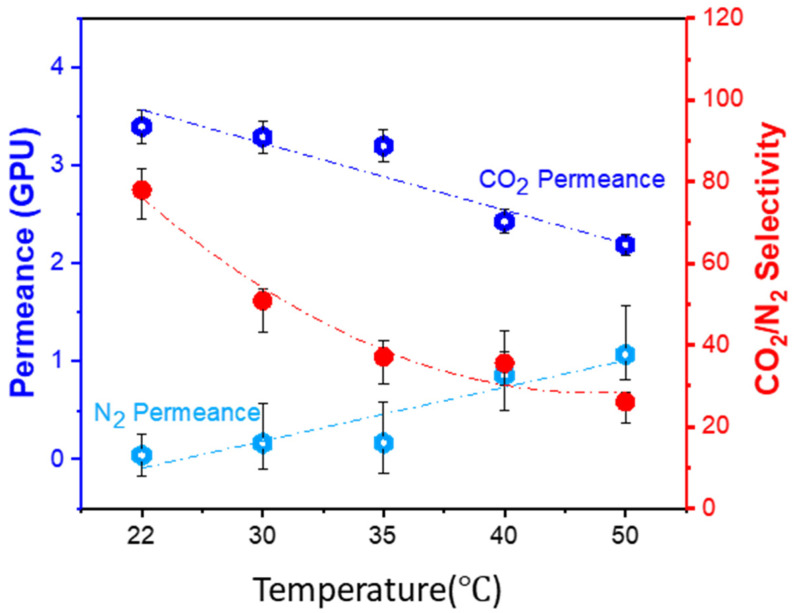
Dependence of the membrane performances on the operating temperature under a feed pressure of 2 bar.

**Figure 9 polymers-15-00124-f009:**
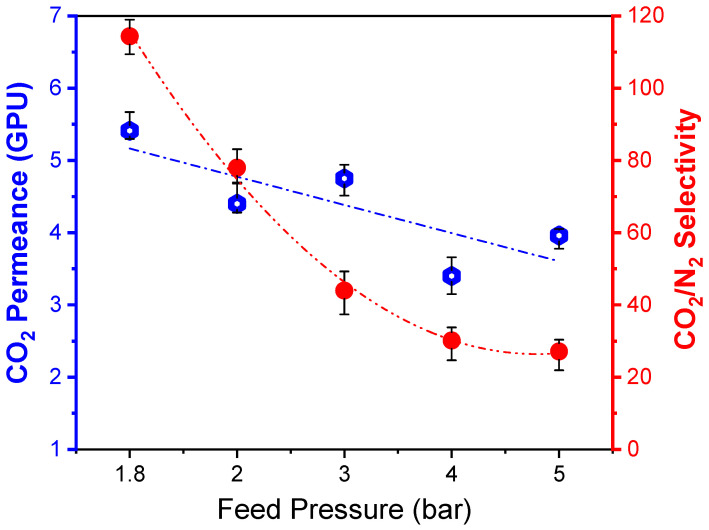
Dependence of the membrane performances on the feed pressure under a constant feed and permeate flow rates of 300 and 20 NmL/min at 25 °C.

## Data Availability

The data presented in this study are available on request from the corresponding author.

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
