# Peer review of "Preparation and Characterization of Polyvinylalcohol/Polysulfone Composite Membranes for Enhanced CO2/N2 Separation"

_polymers, 2022, doi:10.3390/polym15010124_

Round 1

Reviewer 1 Report

Li et al. fabricated a PVA/PSf nanocomposite membranes for CO2/N2 separation. The contribution of PVA enhanced CO2/N2 selectivity of 200% as compared to the pristine PSf membrane. They also studied the effects of operating parameters, and process design and optimization on the CO2-separation which was necessary to explore. I hope this study brought some interesting basic findings in the field of gas separation membranes. My minor comments are sated below:

Page 1,

Lane 2-3: It would be better if the authors write the full forms for PVA and PSF in the article title.  

Lane 15:  Is this selectivity 78.2 obtained from single experiment?

Lane 16: mention the ‘feed pressure’

Lane 40: ‘are usually highly cost’ grammatically incorrect.

Page 3:

Lane 112: mention the temperature for ‘warm tape water’

Lane 118: ‘Placing’ > placing. Please use SI unit for hours.

Lane 119: ‘small thickness of 2 μm’ In the abstract was it 1.7 micron?

Please label glass plate in Figure 1.

Page 6:

Lane 187-201: Need references for all those peaks claim.

Page 7:

No scale bar was shown for Fig. 5 (a-c).

Pages 7-11

Most of these observations and hypothesis were discussed without referring the literatures.

Author Response

Page 1,

Lane 2-3: It would be better if the authors write the full forms for PVA and PSF in the article title.  

Answer: this has been updated in the revised version:  

Preparation and characterization of Polyvinylalcohol/polysulfone composite membranes for enhanced CO2/N2 separation

Lane 15:  Is this selectivity 78.2 obtained from single experiment?

Answer: all the gas separation performance has been obtained from the mixed gas permeation testing, so the selectivity value of 78.2 provided here also from mixed gas permeation experiment.

Lane 16: mention the ‘feed pressure’

Answer: thanks for the comment, the authors have now added the feed pressure from 1.8 to 5 bar.

Lane 40: ‘are usually highly cost’ grammatically incorrect.

Answer: this has been corrected to ‘are usually highly costly’

Page 3:

Lane 112: mention the temperature for ‘warm tape water’

Answer: the temperature for warm tap water of 40-45 °C 

Lane 118: ‘Placing’ > placing. Please use SI unit for hours.

Answer: this has been updated

Lane 119: ‘small thickness of 2 μm’ In the abstract was it 1.7 micron?

Answer: thanks for the review for the comment, this is a mistake, the coating bar we used is 20 μm (for coating solution on top of PSF support), and in the end, we got the thickness of the PVA selective layer of 1.7 μm. this has been corrected in the revised version.

Please label glass plate in Figure 1.

Answer: Figure 1 has been updated by labeling the glass plate.

Page 6:

Lane 187-201: Need references for all those peaks claim.

Answer: the references for the characteristic peaks of PSF and PVA have been included in the revised version

Page 7:

No scale bar was shown for Fig. 5 (a-c).

Answer: the scale bar for Fig. 5 (a-c) has been included in Fig. 5.

Pages 7-11

Most of these observations and hypothesis were discussed without referring the literatures.

Answer: The authors have now cited some references to support our hypothesis and observations, e.g.,

1) It is expected that the gas diffusivity will increase at higher operating temperatures due to kinetic domination (International Journal of Greenhouse Gas Control, 64, 323-332, 2017)

2) the mass transfer coefficient of gas molecules is expected to increase at a higher velocity  (International Journal of Greenhouse Gas Control, 64, 323-332, 2017)

Reviewer 2 Report

This study is focused on the preparation of composite PVA/PSF membranes for gas separation.

1.      Authors named prepared PVA/PSf membranes as nanocomposite. Why? There are two versions used in the work: nanocomposite membrane and composite membranes. In this case, the obtained membranes are composite.

2.      “Therefore, it is crucial to operating this type of membrane in a humidified condition where the polymer chain can be swelled to enhance the gas diffusivity coefficient”.

Humidity is an important parameter, as the authors mentioned in introduction. However, there is no any experiments about the study of the influence of humidity on the composite membrane performance.

3.      Authors claim: “… and thus the CO2/N2 selectivity and CO2 permeability for the water-swollen PVA membranes were significantly improved”. Explain, please, the reason of this.

4.      Explain more detailed why and in what way was SDS used?

The addition of the surfactant of SDS…” addition to what? This is unclear.

5.      It is well known that PVA water solution should be prepared at 90 °C for at least 4 h with stirring. Using escribed preparation conditions may lead to incomplete dissolution of the polymer. In confirmation of this: “The prepared PVA solutions were filtered by 0.8 μm syringe filters to remove any undissolved particles”. As a result, the concentration of PVA solution could be changed.

6.      The PSf membranes were washed using tap water for 1 h. Did the authors used water flow or membranes were soaked? How much time membranes were “gently dried in the atmosphere”?

7.      “The addition of the surfactant of SDS and the removal of the residual glycerol on the PSf surface by washing with warm tape water for at least one hour was conducted to enhance the compatibility between the hydrophobic PSf surface and hydrophilic PVA solution”. Clarify this.

8.      L115-116 “… which ensures the membrane is completely flat to achieve a uniform selective layer without any defects”. Not only this is a prerequisite for the formation of a defect-free selective layer.

9.      L121-122 “followed by a heat treatment at 100 ℃ for 2-3 hours to make crosslinked nanocomposite membranes”. Why it is nanocomposite membrane? Crosslinking of what with what? What is the scheme of the crosslinking reaction?

10.  L141-142. This was mentioned above (L131-132).

11.  L148 “The relative humidity (RH) of feed gas was controlled…”. Where the results of RH measurement?

12.  There is a band 3200-3400 on the FTIR spectrum of pure PSf. This could be the result of the formation of hydrogen bonds between the hydroxyl groups of glycerol or due to residual water. The FTIR spectrum of dry pure PSf membrane without glycerol should be presented.

13.  Add scales to the SEM-microphotographs a,b,c.

What is the reason of such slight increase in PVA layer thickness with increase in polymer concentration from 5 to 7.5 wt.% and such significant increase when using 10 wt.% PVA solution? What was the range of viscosity of PVA solutions?

14.  Authors claim: “Moreover, some PVA polymers might penetrate the PSf matrix to decrease the pore size of the substrate or cause pore blocking and thus leading to the decrease of gas permeance”. Which of PVA polymers couldn’t penetrate the PSf matrix?

15.  P3.4.1. Effect of the PVA concentration. What was the flow feed rate, temperature and pressure in investigation of membrane performance? The performance of pure PSf membrane was studied after it was washing from glycerol or without?

16.  Authors claim: “The membrane performance of the commercial modules is expected to be closer to the results obtained at a low or moderate feed flow”. What is this assumption based on? Have the authors investigated any commercial membrane module under the same conditions?

17.  When studying the effect of feed flow rate, what was the PVA concentration in the solution that was used for membrane preparation?

The results presented in Figure 7 are similar to the pure PSf membrane performance or even worse. Probably, other membrane should be chosen for this experiment.

18.  Authors studied the dependence of the membrane performance on the operation temperature at 2 bar, while in the previous experiment they were used 3 bar. What was the reason to decrease pressure? What was the flow rate in these experiments?

Were the conditions in p.3.4.3 and 3.4.1 the same?

19.  P.3.4.4 What membrane was used to study effect of feed pressure on the membrane performance?

20.  Authors said that “The composite membranes presented better separation performance compared to the pure PSf membranes, which indicates that coating PVA on PSf can effectively improve the membrane selectivity without a significant reduction of gas permeance compared to the pure PSf support membranes”. However, the CO2 permeance was shown to decrease almost in 2-3 times.

21.  L319: “…due to the weakening of coupling transportation between water and CO2 molecules”. To claim this, the authors need to present the humidity values.

22.  Correct the typos:

L11: tap water,

L00: feed pressure.

Author Response

the authors appreciate the reviewer's valuable comments and suggestions, and we have now revised accordingly.

Reviewer 3 Report

In this manuscript, the authors prepared a composite membrane from polyvinyl alcohol (PVA), and polysulfone (PSf), (PVA/PSf) for CO2/N2 separations. The membranes prepared by coating a 7.5 wt.% PVA on top of PSf substrate showed a relatively thin selective layer of 1.7 μm with an enhanced CO2/N2 selectivity of 78.2 which is a ca. 200% increase compared to the pure PSf membranes. The CO2/N2 selectivity decreases at a rapid rate with the increase of feed pressure, while the CO2 permeance shows a slight reduction, which is caused by the weakening of coupling transportation between water and CO2 molecules as well as membrane compaction at higher pressures. Increasing the operating temperature from 22 to 50 °C leads to a slight decrease in CO2 permeance, but a significant reduction in the CO2/N2 selectivity from 78 to 27.1. Moreover, the mass transfer coefficient of gas molecules is expected to increase at a higher velocity, which leads to the increase of CO2 permeance at higher feed flow rates. This research is good for studying the CO2 separation performance of the composite membranes which was significantly dependent on the membrane operating parameters, and process design and optimization are crucial to bringing CO2-separation membranes for Industrial applications in post-combustion carbon capture. The interpretations of the results are well discussed. The quantity and quality of the figures are appropriate. We believe that this research subject is promising for developing composite membranes with high performance for gas separation.  

Summary: I recommend publishing this manuscript after considering my comments on the attached file.

Author Response

the authors thank your comments, and we have now revised accordingly.

Round 2

Reviewer 2 Report

The authors partly corrected the manuscript, however there are some comment that need to be explained more clearly.

I would recommend this paper to publication after major revision.

1.      The authors should measure the swelling degree of prepared composite PVA/PSf membranes.

2.      The scale bars of SEM microphotographs are still unclear. The font should be increased.

3.      The authors answer: this might be the misunderstanding, we have now revised the sentence: “Moreover, part of PVA polymers might penetrate the PSf matrix to decrease the pore size of the substrate or cause pore blocking and thus leading to the decrease of gas permeance”.

In this case it is better to use “lower molecular weight PVA fractions might penetrate…”, if authors mean this.

4.      So far as authors tested some pilot modules and claimed “The membrane performance of the commercial modules is expected to be closer to the results obtained at a low or moderate feed flow”, some results of previous work should be added as well as the reference.

5.      Authors responded: for the variation of feed flow, as we used a back pressure controller, and it is difficult to get higher flow when the testing pressure is 2 bar, that’s why we choose to test at 3bar-this is the reason higher feed flow of 500Nml/min has not been conducted in this work, and we also described that “…much higher feed flow rates of >600 Nml/min have not been conducted in this work" in section 3.4.2. while for other experiment, as 2 bar of feed pressure Is widely used, and we do not want to operate membrane system at higher pressure in the real application due to a significant Increase of power demand from feed flue gas compression.

Authors should add the explanation to the manuscript.

6.      The results in figures 6 and 9 do not correlate. In figure 6 CO2 permeance (GPU) is ~3.3 for membrane from 7.5 wt.% PVA, but in Figure 9 it is 4.4 at the same conditions, while the selectivity is similar. Explain this.

The authors partly corrected the manuscript, however there are some comment that need to be explained more clearly.

I would recommend this paper to publication after major revision.

1.      The authors should measure the swelling degree of prepared composite PVA/PSf membranes.

2.      The scale bars of SEM microphotographs are still unclear. The font should be increased.

3.      The authors answer: this might be the misunderstanding, we have now revised the sentence: “Moreover, part of PVA polymers might penetrate the PSf matrix to decrease the pore size of the substrate or cause pore blocking and thus leading to the decrease of gas permeance”.

In this case it is better to use “lower molecular weight PVA fractions might penetrate…”, if authors mean this.

4.      So far as authors tested some pilot modules and claimed “The membrane performance of the commercial modules is expected to be closer to the results obtained at a low or moderate feed flow”, some results of previous work should be added as well as the reference.

5.      Authors responded: for the variation of feed flow, as we used a back pressure controller, and it is difficult to get higher flow when the testing pressure is 2 bar, that’s why we choose to test at 3bar-this is the reason higher feed flow of 500Nml/min has not been conducted in this work, and we also described that “…much higher feed flow rates of >600 Nml/min have not been conducted in this work" in section 3.4.2. while for other experiment, as 2 bar of feed pressure Is widely used, and we do not want to operate membrane system at higher pressure in the real application due to a significant Increase of power demand from feed flue gas compression.

Authors should add the explanation to the manuscript.

6.      The results in figures 6 and 9 do not correlate. In figure 6 CO2 permeance (GPU) is ~3.3 for membrane from 7.5 wt.% PVA, but in Figure 9 it is 4.4 at the same conditions, while the selectivity is similar. Explain this.

The authors partly corrected the manuscript, however there are some comment that need to be explained more clearly.

I would recommend this paper to publication after major revision.

1.      The authors should measure the swelling degree of prepared composite PVA/PSf membranes.

2.      The scale bars of SEM microphotographs are still unclear. The font should be increased.

3.      The authors answer: this might be the misunderstanding, we have now revised the sentence: “Moreover, part of PVA polymers might penetrate the PSf matrix to decrease the pore size of the substrate or cause pore blocking and thus leading to the decrease of gas permeance”.

In this case it is better to use “lower molecular weight PVA fractions might penetrate…”, if authors mean this.

4.      So far as authors tested some pilot modules and claimed “The membrane performance of the commercial modules is expected to be closer to the results obtained at a low or moderate feed flow”, some results of previous work should be added as well as the reference.

5.      Authors responded: for the variation of feed flow, as we used a back pressure controller, and it is difficult to get higher flow when the testing pressure is 2 bar, that’s why we choose to test at 3bar-this is the reason higher feed flow of 500Nml/min has not been conducted in this work, and we also described that “…much higher feed flow rates of >600 Nml/min have not been conducted in this work" in section 3.4.2. while for other experiment, as 2 bar of feed pressure Is widely used, and we do not want to operate membrane system at higher pressure in the real application due to a significant Increase of power demand from feed flue gas compression.

Authors should add the explanation to the manuscript.

6.      The results in figures 6 and 9 do not correlate. In figure 6 CO2 permeance (GPU) is ~3.3 for membrane from 7.5 wt.% PVA, but in Figure 9 it is 4.4 at the same conditions, while the selectivity is similar. Explain this.

The authors partly corrected the manuscript, however there are some comment that need to be explained more clearly.

I would recommend this paper to publication after major revision.

1.      The authors should measure the swelling degree of prepared composite PVA/PSf membranes.

2.      The scale bars of SEM microphotographs are still unclear. The font should be increased.

3.      The authors answer: this might be the misunderstanding, we have now revised the sentence: “Moreover, part of PVA polymers might penetrate the PSf matrix to decrease the pore size of the substrate or cause pore blocking and thus leading to the decrease of gas permeance”.

In this case it is better to use “lower molecular weight PVA fractions might penetrate…”, if authors mean this.

4.      So far as authors tested some pilot modules and claimed “The membrane performance of the commercial modules is expected to be closer to the results obtained at a low or moderate feed flow”, some results of previous work should be added as well as the reference.

5.      Authors responded: for the variation of feed flow, as we used a back pressure controller, and it is difficult to get higher flow when the testing pressure is 2 bar, that’s why we choose to test at 3bar-this is the reason higher feed flow of 500Nml/min has not been conducted in this work, and we also described that “…much higher feed flow rates of >600 Nml/min have not been conducted in this work" in section 3.4.2. while for other experiment, as 2 bar of feed pressure Is widely used, and we do not want to operate membrane system at higher pressure in the real application due to a significant Increase of power demand from feed flue gas compression.

Authors should add the explanation to the manuscript.

6.      The results in figures 6 and 9 do not correlate. In figure 6 CO2 permeance (GPU) is ~3.3 for membrane from 7.5 wt.% PVA, but in Figure 9 it is 4.4 at the same conditions, while the selectivity is similar. Explain this.

The authors partly corrected the manuscript, however there are some comment that need to be explained more clearly.

I would recommend this paper to publication after major revision.

1.      The authors should measure the swelling degree of prepared composite PVA/PSf membranes.

2.      The scale bars of SEM microphotographs are still unclear. The font should be increased.

3.      The authors answer: this might be the misunderstanding, we have now revised the sentence: “Moreover, part of PVA polymers might penetrate the PSf matrix to decrease the pore size of the substrate or cause pore blocking and thus leading to the decrease of gas permeance”.

In this case it is better to use “lower molecular weight PVA fractions might penetrate…”, if authors mean this.

4.      So far as authors tested some pilot modules and claimed “The membrane performance of the commercial modules is expected to be closer to the results obtained at a low or moderate feed flow”, some results of previous work should be added as well as the reference.

5.      Authors responded: for the variation of feed flow, as we used a back pressure controller, and it is difficult to get higher flow when the testing pressure is 2 bar, that’s why we choose to test at 3bar-this is the reason higher feed flow of 500Nml/min has not been conducted in this work, and we also described that “…much higher feed flow rates of >600 Nml/min have not been conducted in this work" in section 3.4.2. while for other experiment, as 2 bar of feed pressure Is widely used, and we do not want to operate membrane system at higher pressure in the real application due to a significant Increase of power demand from feed flue gas compression.

Authors should add the explanation to the manuscript.

6.      The results in figures 6 and 9 do not correlate. In figure 6 CO2 permeance (GPU) is ~3.3 for membrane from 7.5 wt.% PVA, but in Figure 9 it is 4.4 at the same conditions, while the selectivity is similar. Explain this.

The authors partly corrected the manuscript, however there are some comment that need to be explained more clearly.

I would recommend this paper to publication after major revision.

1.      The authors should measure the swelling degree of prepared composite PVA/PSf membranes.

2.      The scale bars of SEM microphotographs are still unclear. The font should be increased.

3.      The authors answer: this might be the misunderstanding, we have now revised the sentence: “Moreover, part of PVA polymers might penetrate the PSf matrix to decrease the pore size of the substrate or cause pore blocking and thus leading to the decrease of gas permeance”.

In this case it is better to use “lower molecular weight PVA fractions might penetrate…”, if authors mean this.

4.      So far as authors tested some pilot modules and claimed “The membrane performance of the commercial modules is expected to be closer to the results obtained at a low or moderate feed flow”, some results of previous work should be added as well as the reference.

5.      Authors responded: for the variation of feed flow, as we used a back pressure controller, and it is difficult to get higher flow when the testing pressure is 2 bar, that’s why we choose to test at 3bar-this is the reason higher feed flow of 500Nml/min has not been conducted in this work, and we also described that “…much higher feed flow rates of >600 Nml/min have not been conducted in this work" in section 3.4.2. while for other experiment, as 2 bar of feed pressure Is widely used, and we do not want to operate membrane system at higher pressure in the real application due to a significant Increase of power demand from feed flue gas compression.

Authors should add the explanation to the manuscript.

6.      The results in figures 6 and 9 do not correlate. In figure 6 CO2 permeance (GPU) is ~3.3 for membrane from 7.5 wt.% PVA, but in Figure 9 it is 4.4 at the same conditions, while the selectivity is similar. Explain this.

The authors partly corrected the manuscript, however there are some comment that need to be explained more clearly.

I would recommend this paper to publication after major revision.

1.      The authors should measure the swelling degree of prepared composite PVA/PSf membranes.

2.      The scale bars of SEM microphotographs are still unclear. The font should be increased.

3.      The authors answer: this might be the misunderstanding, we have now revised the sentence: “Moreover, part of PVA polymers might penetrate the PSf matrix to decrease the pore size of the substrate or cause pore blocking and thus leading to the decrease of gas permeance”.

In this case it is better to use “lower molecular weight PVA fractions might penetrate…”, if authors mean this.

4.      So far as authors tested some pilot modules and claimed “The membrane performance of the commercial modules is expected to be closer to the results obtained at a low or moderate feed flow”, some results of previous work should be added as well as the reference.

5.      Authors responded: for the variation of feed flow, as we used a back pressure controller, and it is difficult to get higher flow when the testing pressure is 2 bar, that’s why we choose to test at 3bar-this is the reason higher feed flow of 500Nml/min has not been conducted in this work, and we also described that “…much higher feed flow rates of >600 Nml/min have not been conducted in this work" in section 3.4.2. while for other experiment, as 2 bar of feed pressure Is widely used, and we do not want to operate membrane system at higher pressure in the real application due to a significant Increase of power demand from feed flue gas compression.

Authors should add the explanation to the manuscript.

6.      The results in figures 6 and 9 do not correlate. In figure 6 CO2 permeance (GPU) is ~3.3 for membrane from 7.5 wt.% PVA, but in Figure 9 it is 4.4 at the same conditions, while the selectivity is similar. Explain this.

The authors partly corrected the manuscript, however there are some comment that need to be explained more clearly.

I would recommend this paper to publication after major revision.

1.      The authors should measure the swelling degree of prepared composite PVA/PSf membranes.

2.      The scale bars of SEM microphotographs are still unclear. The font should be increased.

3.      The authors answer: this might be the misunderstanding, we have now revised the sentence: “Moreover, part of PVA polymers might penetrate the PSf matrix to decrease the pore size of the substrate or cause pore blocking and thus leading to the decrease of gas permeance”.

In this case it is better to use “lower molecular weight PVA fractions might penetrate…”, if authors mean this.

4.      So far as authors tested some pilot modules and claimed “The membrane performance of the commercial modules is expected to be closer to the results obtained at a low or moderate feed flow”, some results of previous work should be added as well as the reference.

5.      Authors responded: for the variation of feed flow, as we used a back pressure controller, and it is difficult to get higher flow when the testing pressure is 2 bar, that’s why we choose to test at 3bar-this is the reason higher feed flow of 500Nml/min has not been conducted in this work, and we also described that “…much higher feed flow rates of >600 Nml/min have not been conducted in this work" in section 3.4.2. while for other experiment, as 2 bar of feed pressure Is widely used, and we do not want to operate membrane system at higher pressure in the real application due to a significant Increase of power demand from feed flue gas compression.

Authors should add the explanation to the manuscript.

6.      The results in figures 6 and 9 do not correlate. In figure 6 CO2 permeance (GPU) is ~3.3 for membrane from 7.5 wt.% PVA, but in Figure 9 it is 4.4 at the same conditions, while the selectivity is similar. Explain this.

Author Response

  1. The authors should measure the swelling degree of prepared composite PVA/PSf membranes

Answer:  we did measure the swelling degree of the prepared composite PVA/PSf membranes. to avoid any misunderstanding, we have now described it more clearly in line 135 "The swelling degree (SD) of the PVA/PSf composite  membranes was evaluated... "

2. The scale bars of SEM microphotographs are still unclear. The font should be increased.

Answer:  the SEM images of Fig. 5 have been updated, and the scale bar is clearly shown in the revised version.

3. The authors answer: this might be the misunderstanding, we have now revised the sentence: “Moreover, part of PVA polymers might penetrate the PSf matrix to decrease the pore size of the substrate or cause pore blocking and thus leading to the decrease of gas permeance”.

In this case it is better to use “lower molecular weight PVA fractions might penetrate…”, if authors mean this.

Answer: thanks for the comment, this has now been revised according to your suggestion.

  1. So far as authors tested some pilot modules and claimed “The membrane performance of the commercial modules is expected to be closer to the results obtained at a low or moderate feed flow”, some results of previous work should be added as well as the reference.

Answer: for this membrane, we have not tested any commercial module.  therefore no such results can be included. the statement in line 254 has now been changed to t " It is expected that the commercial modules should be operated at a low or moderate feed flow" to avoid any misunderstanding.

  1. Authors responded: “for the variation of feed flow, as we used a back pressure controller, and it is difficult to get higher flow when the testing pressure is 2 bar, that’s why we choose to test at 3bar-this is the reason higher feed flow of 500Nml/min has not been conducted in this work, and we also described that “…much higher feed flow rates of >600 Nml/min have not been conducted in this work" in section 3.4.2. while for other experiment, as 2 bar of feed pressure Is widely used, and we do not want to operate membrane system at higher pressure in the real application due to a significant Increase of power demand from feed flue gas compression”.

Authors should add the explanation to the manuscript.

Answer: we have now included this explanation in the revised version.

line 246 "It should be noted that it is difficult to get the higher flow rate of >300Nml/min when the feed pressure is controlled at 2 bar by a back pressure controller. Therefore the feed pressure of 3bar was applied for the variation of the feed flow rate"

line 156: "It is worth noting that a feed pressure of 2 bar is widely used in real-life applications due to the expected significantly increased power demand for the flue gas compression to be operated at a higher feed pressure. Therefore most testing experiments were conducted at the feed pressure of 2 bar except for the influence of feed flow rate where a feed pressure of 3 bar was used."

  1. The results in figures 6 and 9 do not correlate. In figure 6 CO2 permeance (GPU) is ~3.3 for membrane from 7.5 wt.% PVA, but in Figure 9 it is 4.4 at the same conditions, while the selectivity is similar. Explain this.

answer: the authors have checked carefully the original data, and it was found the experimental data in Fig. 6 for 7.5 wt% is wrong. we appreciate the reviewer's comment, and Fig. 6 has now been updated.
